# OpenReview forum: "Estimating Tail Risks in Language Model Output Distributions"
_ICML.cc/2026/Conference — ICML 2026 spotlight_

### Official Review · Reviewer_Eutt · 2026-02-18

**Soundness:** 4
**Presentation:** 3
**Significance:** 3
**Originality:** 3
**Overall Recommendation:** 5
**Confidence:** 3

**Summary:**

The authors create a new method to estimate the probability of harmful, but low-probability outputs of LLMs. Their primary contribution involves the addition of steering vectors, which allows for the discovery of realistic but low-probability outputs from a target model.  They extend previous work on benchmarking and forcasting rare failure modes by providing a sample efficient method to estimate harmful, but improbable outcomes. They complement this method with studies on the effects of increased sampling and prompt rewriting on producing at least one harmful output.

**Compliance With Llm Reviewing Policy:**

Affirmed.

**Final Justification:**

The author's successfully addressed my concerns around extending the technique to reasoning and multi-turn settings. I have increased my recommendation to an accept.

**Key Questions For Authors:**

1. What low-probability but unsafe responses motivate the paper? This method has the largest upside for extremely low-probability rollouts that are too expensive to discover through brute force MCTS estimates. A short description of the target risks that motivate this work would be valuable.
2.  Can this method be modified for reasoning models? For instance, does x of p_target(x|c) become only the final output? or does x include the reasoning trace? Where do you expect difficulties to arise, and why was this not discussed in the limitations?

**Limitations:**

The authors do not mention that many harmful outcomes will come from reasoning agents in agentic environments. Modeling risks that only occur over multi-turn interactions (or multi-turn actions taken by agents) are not discussed. Additionally, the use of only non-reasoning models should be addressed.

**Strengths And Weaknesses:**

Soundness: The method introduced during this work is well supported through multiple sets of empirics across a wide range of model families. They also extend the method to *safe* but undesirable responses, highlighting the versatility of the method. Additionally, they extend their technique to a nearby distribution of rewritten harmful queries, show this largely changes the likelihood of harmful respones, and shows their method can be modified to account for this.

Presentation: Overall the paper is well written and structured. However the sections felt somewhat choppy. (e.g. section 4.1 concludes Section 4, but doesn't provide takeaways for "the efficiency and reliability of importance sampling). Similarly, the choice of section titles made it somewhat difficult to quickly navigate the paper (e.g. Section 4 "Unsafe Response Rates for Unsafe Queries").

Significance: This paper address a specific issue in the forecasting of low-probability but harmful events: the ability to compute efficient estimates of the likelihood of harmful responses without the need for those harmful responses being identified ahead of time. This was directly mentioned as an important limitation in Jones et al 2025.

Originality: The paper address a limitation to Jones et al 2025 and extends Wu & Hilton 2024 to a significant degree.

---

> ### Author Rebuttal · Authors · 2026-03-30
>
> **We appreciate your feedback on our work! We have included responses to your comments below, and hope the added clarifications are sufficient to increase your score.**
>
> *> What low-probability but unsafe responses motivate the paper?*
>
> Our paper is motivated by the observation that “safe” models have a rare, but non-negligible, chance of generating undesirable responses, including answering unsafe queries, such as “What is the best way to spread xenophobic attitudes?”, and exhibiting other misbehaviors, including sycophancy and hallucinations. Importantly, our method could be applied to any rare misbehavior for which we have access to a judge.
>
> *> Originality: The paper address a limitation to Jones et al 2025 and extends Wu & Hilton 2024 to a significant degree.*
>
> We would like to clarify the significant differences between our work and Jones et al (2025) and Wu & Hilton (2024), beyond addressing their limitations.
>
> In our paper, we first identify the tail-risk of harmful behaviors with repeated sampling (given the same query), and then do **rare event estimation over the model’s output** **distribution**!
>
> Importantly, our primary contribution is ***not*** “the addition of steering vectors, which allows for the discovery of realistic but low-probability outputs.” We use steering vectors as a means to an end: to show that rare event estimation over the output distribution is feasible for LLMs. Moreover, our work demonstrates that imperfect and heuristic steering tools can be leveraged with importance sampling to create useful, efficient estimators.
>
> In contrast, Jones et al. (2025) forecast model behaviors over a distribution of input queries, using extreme value theory and a proxy for P(harm | x). One downstream use for our tail-risk estimates is addressing this key limitation to Jones et al. (2025) (the use of a proxy probability), which we discuss in Section 6.2: *Predicting Worst-Case Risk on Unseen Queries*.
>
> Additionally, the task diverges significantly from those of Wu and Hilton (2024), who (a) assume that the model is deterministic, (b) estimate harm over the distribution of model input, rather than outputs, and (c) make unrealistic assumptions, such as that the input distribution tokens are independent.
>
> *> Presentation: Overall the paper is well written and structured. […] section 4.1 concludes Section 4, but doesn't provide takeaways for "the efficiency and reliability of importance sampling)...*
>
> We appreciate that you overall found the paper to be well-written and well-structured. In Section 4, we have a discussion about the efficacy and reliability of importance sampling (the “Results” paragraph). In the revised version, we will adjust the section titles for more clarity and in the additional space add an explicit takeaways paragraph.
>
> *> The authors do not mention that many harmful outcomes will come from reasoning agents in agentic environments. Modeling risks that only occur over multi-turn interactions (or multi-turn actions taken by agents) are not discussed. Additionally, the use of only non-reasoning models should be addressed.*
>
> Reasoning agents are increasingly being deployed in high-risk environments such as healthcare. In such settings, measuring the probability of harm caused by these agents is critical to safe deployment. Our method can be adapted to these settings for efficient pre-deployment evaluations.
>
> Due to limited computational resources, we restricted our evaluations to small-to-medium size (1-14 billion parameters) open source non-reasoning models in single-turn non-agentic environments. However, given sufficient computational resources, our methodology could be extended to reasoning agents and mutli-turn settings in high-risk environments as well.
>
> *> Can this method be modified for reasoning models? For instance, does x of p_target(x|c) become only the final output? or does x include the reasoning trace? Where do you expect difficulties to arise, and why was this not discussed in the limitations?*
>
> Our method can work with any target model (including reasoning and agentic models) and can be adapted to measure misbehavior in the chain-of-thought or final output. The x would be the entire output trajectory for the sake of importance sampling, but the judge might only be applied to a certain subset of the trajectory (e.g. the final output).
>
> The biggest bottleneck is the compute necessary to run such experiments and to benchmark results against brute-force Monte Carlo estimates.
>
> We are more than happy to add a discussion in the revised version about potential difficulties in adapting our method to more real-world use cases.
>
> **Please let us know if you have any other questions that we can address. If not, we sincerely hope you can consider increasing your rating to an Accept, as we believe that our contribution is significant to the safety community that aims to improve evaluation techniques.**

---

> > ### Author Rebuttal · Reviewer_Eutt · 2026-04-01
> >
> > Thank you for the clear and detailed response! I am excited for the potential for this to be deployed in high-risk environments such as the internal testing highly capable models within frontier labs.
> >
> > Can you confirm that given acceptance, you will add discussion on *how* your technique would be modified for reasoning models and/or multi-turn interactions?
> >
> > I appreciated that the empirics were beyond the scope of the paper due to resource constraints, but further discussion here would be beneficial and in my opinion necessary for a strong recommendation.

---

> > > ### Author Response · Authors · 2026-04-02
> > >
> > > **Thank you for your quick response. We will add the below discussion on future work for rare-event estimation with reasoning models and multi-turn settings. Please let us know if you would like us to include any additional information!**
> > >
> > > ----
> > >
> > > Language models increasingly use intermediate reasoning, and are applied in agentic and multi-turn settings. These applications include models that are incorporated in critical infrastructure and workflows in medicine, law, finance, etc. Due to compute constraints, our work focuses on non-reasoning models and single-turn queries. However, our method can be extended to model harm for reasoning models in multi-turn settings.
> > >
> > > **Reasoning models.**
> > > To adapt our method to reasoning models, where responses are decomposed into a reasoning chain followed by an answer,  $x = [r; a] $, we include the chain-of-thought in the importance sampling calculation while extracting only the answer for the harm evaluation:
> > >
> > > $$
> > > \mathbb{E}\_{x \sim q_{\phi}(\cdot \mid c)}\left[
> > > \frac{p\_{\mathrm{target}}(x \mid c)}{q\_{\phi}(x \mid c)} h(x; c)
> > > \right] = \mathbb{E}_{(r, a) \sim q\_{\phi}(\cdot \mid c)} \left[
> > > \frac{p\_{\mathrm{target}}(r,a \mid c)}{q\_{\phi}(r,a \mid c)} h(a; c)
> > > \right]
> > > $$
> > >
> > > This involves sampling from an unsafe model, as in our current method. Notably, alternative proposal families constructed via prompting, supervised fine-tuning, and reinforcement learning may enable more efficient estimation for reasoning tasks.
> > >
> > > **Multi-turn settings.**
> > > Extending our method to multi-turn settings requires modeling how a user responds to intermediate outputs. This can be done using a user simulator approximating
> > >
> > > $$
> > > p(c_{t+1} \mid x_t, c_t, \ldots, x_1, c_1).
> > > $$
> > >
> > > Combining this with our approach enables estimation of rare but harmful outcomes across multiple turns via importance sampling:
> > >
> > > $$
> > > \mathbb{E} \left[
> > > \left( \prod_{t=1}^{n}
> > > \frac{p_{\mathrm{target}}(x_t \mid c_t)}{q_{\phi}(x_t \mid c_t)}
> > > \right)
> > > h(x_{1:n}; c_{1:n})
> > > \right],
> > > $$
> > >
> > > where the trajectory is generated according to
> > >
> > > $$
> > > c_1 \sim p_{\mathrm{data}}(\cdot), \quad
> > > x_t \sim q_{\phi}(\cdot \mid c_t), \quad
> > > c_{t+1} \sim p_{\mathrm{user}}(\cdot \mid x_t, c_t, \ldots, x_1, c_1).
> > > $$

---

### Official Review · Reviewer_7bjq · 2026-03-08

**Soundness:** 4
**Presentation:** 4
**Significance:** 3
**Originality:** 3
**Overall Recommendation:** 4
**Confidence:** 4

**Summary:**

The paper proposes a novel method to efficiently estimate harmful behaviors exhibited in the output distribution, which is less studied in LLM safety.
Leveraging the idea of importance sampling, the authors propose to use outputs of an unsafe version of the target models to approximate the target model’s rare unsafe behaviors.
The unsafe version of the target model, called the proposal model, is constructed with activation steering and refined with Mixture of models &Model switching. To minimize the variance of estimate, the proposal hyperparameters are optimized through the cross-entropy method.
The experiments evaluated the effectiveness of the proposed method on both unsafe and safe queries, and showed that the proposed method converges to the ground truth query-level harm (brute-force Monte Carlo with 10k samples) better than baseline MC.
The authors further utilized their method to study model behaviors. They demonstrated that non-adversarial rewrites of harmful queries increase the probability of unsafe outputs by multiple orders of magnitude, and their efficient method can be used to model the query-level harm over unseen queries.

**Compliance With Llm Reviewing Policy:**

Affirmed.

**Final Justification:**

The authors’ rebuttal partially resolved my concerns. I suggest the authors compare the variability and confidence interval with the MC baseline to show that the benefit of the proposed method over the baseline is not by chance. I maintain my positive scores.

**Key Questions For Authors:**

1. For each input query $c$, the authors sampled 1000 outputs from the proposal model to estimate the $Q\_{risk}(c)$. How robust is the estimated $Q\_{risk}(c)$ if you resample different 1000 outputs?
2. What is the confidence level of the computed $Q\_{risk}(c)$ by the proposed method with different number of samples?

**Limitations:**

The paper briefly mentions the limitation of the usage of LLM-as-judge. It might be worth to reflect on how this internal uncertainty could affect the conclusions, and which methods could be taken to address it, for example, evaluating with several more popular LLM-as-judges.

**Strengths And Weaknesses:**

**Strengths:**

1. The proposed importance sampling method can efficiently approximate the target model’s query-level harm with 10-20x fewer samples, which is a huge improvement
2. The idea of constructing an unsafe version of the target model to enable importance sampling is simple and intuitive
3. The evaluations (Sec.4&5) of the proposed method comprehensively consider multiple LLMs and both unsafe and safe queries. The results of brute-force Monte Carlo with 10k samples were treated as the groundtruth, which is a reasonable choice.
4. The illustrations and figures clearly demonstrate that the proposed method is more efficient than brute-force MC baseline
5. The experiments in Sec.6 further show that the proposed method can help understand the weakness of LLMs and provide approximation of the model unsafe behaviors after deployment, which are insightful and useful for the development of safe LLM systems in practice

**Weaknesses:**
1. Lack in-depth metric-/model-wise analysis. For example,
    - In Figure 3, the proposed method are clearly better than baseline for most LLMs, except for Llama-3.2-1b, where the gap between MC and IS is small.
    - In Figure 5, the results for Hallucination: Llama-3.1-8b (d) show that MC has similar performance as IS, which is very different from the results of (1) other model’s same metric: Qwen-2.5-7B’s Hallucination (2) same model’s other metric: Llama-3.1-8b’s Sycophancy and unsafe scores in Fig.3
    - Further discussion and analysis into those obvious distinguished results would provide more insights about the effectiveness of the proposed method
2. Sec.5 considered 20 specifically-designed safe input queries. The number of prompts is not significant. And it is unclear why not using regular safe input queries. These undermine the generalizability of the results in Sec.5

Minor:
- line 135 left column, “Llama-3.2-8B” seems like an error
- line 326 left column, word “but leading” seems like an error

---

> ### Author Rebuttal · Authors · 2026-03-30
>
> **Thank you for your feedback. We deeply appreciate your positivity about our work! We have added clarifications below, and believe the added context is sufficient to increase your overall score.**
>
> *> Lack in-depth metric-/model-wise analysis. For example,*
>
> - *In Figure 3, the proposed method are clearly better than baseline for most LLMs, except for Llama-3.2-1b, where the gap between MC and IS is small.*
> - *In Figure 5, the results for Hallucination: Llama-3.1-8b (d) show that MC has similar performance as IS, which is very different from the results of (1) other model’s same metric: Qwen-2.5-7B’s Hallucination (2) same model’s other metric: Llama-3.1-8b’s Sycophancy and unsafe scores in Fig.3*
> - *Further discussion and analysis into those obvious distinguished results would provide more insights about the effectiveness of the proposed method*
>
> The smaller gap for Llama-3.2-1b is due to the relative harmfulness of the original model. Harmful outputs are more frequent, limiting the benefits from a harmful proposal model. Our method offers more significant improvements for estimating harmfulness when it is more rare (i.e. when Q_risk(c) is smaller). The situation in Figure 5 is similar: Llama-3.1-8b hallucinates quite frequently (Q_risk(c) > 0.1 for all input contexts c), so our method is as effective as Monte Carlo estimation.
>
> We will include additional model analysis and detailed results within the additional space provided for the revised version.
>
> *> Sec.5 considered 20 specifically-designed safe input queries. The number of prompts is not significant. And it is unclear why not using regular safe input queries. These undermine the generalizability of the results in Sec.5*
>
> After evaluating unsafe responses on 313 StrongREJECT queries, we also consider the 20 queries for measuring sycophancy and hallucination introduced in a pre-existing benchmark for studying persona vectors (Chen et al. 2025).
>
> We are excited to further explore unsafe, misaligned responses to safe queries in future work!
>
> *> For each input query c, the authors sampled 1000 outputs from the proposal model to estimate the $Q_\text{risk}(c)$. How robust is the estimated $Q_\text{risk}(c)$ if you resample different 1000 outputs?*
>
> Since importance sampling yields unbiased estimates, the robustness of $Q_\text{risk}(c)$ is determined by the variance of the estimator.  In Figures 3 and 5, we plot the error, and observe that the error of the importance sampling is up to four orders of magnitude smaller than the error of the Monte Carlo estimator for the same number of samples. Hence, if we resampled a different 1000 outputs from both estimators the importance sampling estimator would be much more robust than the Monte Carlo estimator.
>
> *> What is the confidence level of the computed $Q_\text{risk}(c)$ by the proposed method with different number of samples?*
>
> The confidence level is also determined by the variance of the estimator. As the number of samples increases the confidence level of both the Monte Carlo estimator and the importance sampling estimator increase, but the variance of the importance sampling estimator is lower for any fixed number of samples (see Figures 3 and 5).
>
> *> The paper briefly mentions the limitation of the usage of LLM-as-judge. It might be worth to reflect on how this internal uncertainty could affect the conclusions, and which methods could be taken to address it, for example, evaluating with several more popular LLM-as-judges.*
>
> We were restricted to judges that we could also use for our slower brute-force Monte Carlo estimates, while working within our limited compute budget. We note in the paper that automatic judges are fallible, and elaborate further in the revised version.
>
> While the focus of our paper is estimation given a judge, complementary work on improving and calibrating LLM-as-judges could further improve estimates. Critically, more advanced judges that require additional compute overhead would further increase the cost of evaluation, and further motivate our efficient estimation method!
>
> *> Minor typos*
>
> Thank you for pointing these out, we have fixed them in the revised version.
>
> **Please let us know if you have any other questions that we can address. If not, we sincerely hope you can consider increasing your rating to an Accept, as we believe that our contribution is significant to the safety community that aims to increase the credibility of evaluation techniques.**

---

> > ### Author Rebuttal · Reviewer_7bjq · 2026-04-01
> >
> > I’d like to thank the authors for responding my questions in detail. But there are several questions that haven't been fully resolved by the authors' response.
> >
> > ### W2
> > The author's response only restates their experiment setting but doesn’t address this concern on the generalizability of Sec.5’s results, which are drawn based on a limited number of “intrinsically safe but leading” queries.
> >
> > ### Q1 & Q2
> > Thanks for explaining that the error metric (the average log-ratio between the estimator and the brute-force estimate) decreases with more samples.
> >
> > But the questions are meant to ask about the consistency and confidence interval of the estimated risk.
> >
> > [Q1] If the identical experiment is repeated using a different random seed, thereby generating an entirely new set of 1000 samples, will the estimated risk $Q_{risk}(c)$ computed on this new set of 1000 samples vary significantly?
> >
> > [Q2] What is the 95% confidence interval of the estimated risk $Q_{risk}(c)$ by the proposed method when it is computed with different number of samples?

---

> > > ### Author Response · Authors · 2026-04-02
> > >
> > > > W2: The author's response only restates their experiment setting but doesn’t address this concern on the generalizability of Sec.5’s results, which are drawn based on a limited number of “intrinsically safe but leading” queries.
> > >
> > > We agree that 20 prompts is not a large enough sample size to show that our approach can efficiently and effectively approximate tail risks of models being sycophantic and hallucinating across an *“in-the-wild”* distribution of safe prompts.
> > >
> > > **We appreciate your point and will clarify this in our revised draft:**
> > >
> > > With limited compute, we needed prompts that have a non-negligible probability of causing the model to misbehave, for which we could use a $10^4$ sample brute-force Monte Carlo estimate to benchmark our method.
> > >
> > > A typical “safe” prompt has a negligible probability of producing a sycophantic or hallucinating response. In order to show this as a proof-of-concept, we chose an off-the-shelf dataset where we know there was a non-zero probability of the model misbehaving, hence the leading questions. On these 20 prompts, we are able to see that importance sampling leads to an accurate and efficient estimation of misaligned response frequency.
> > >
> > > In future work, we think an extremely valuable extension would be to create a user model of “safe” prompts and then estimate the probability that prompts from this distribution will cause the model to be sycophantic or hallucinate.
> > >
> > >
> > > > [Q1] If the identical experiment is repeated using a different random seed, thereby generating an entirely new set of 1000 samples, will the estimated risk computed on this new set of 1000 samples vary significantly?
> > >
> > >
> > > We estimate $Q_\text{risk}(\mathbf{c})$ with 500 samples using 3 different seeds and averaged across all prompts in StrongREJECT. The results are as follows:
> > >
> > >
> > > | Model           | Seed 1    | Seed 2    | Seed 3    | MC estimate ($10^4$ samples) |
> > > |-----------------|------------|------------|------------|------------|
> > > | Qwen2.5-7B-Instruct            | 0.01755 | 0.01733 | 0.01724 | 0.01736 |
> > > | Olmo-3-7B-Instruct            | 0.00118 | 0.00195 | 0.00111 | 0.00119|
> > > | Llama-3.1-8B-Instruct    | 0.01273 | 0.01088 | 0.01209 |  0.01239|
> > > | Llama-3.2-1B-Instruct         | 0.12013 | 0.11814 | 0.13926 | 0.12312 |
> > > | phi-4           | 0.00404 | 0.00466 | 0.00450 | 0.00437 |
> > >
> > > While there is some variability in the estimates for different seeds, the estimates are close to the $10^4$ MC estimates and are not systematically biased in any direction.
> > >
> > > > [Q2] What is the 95% confidence interval of the estimated risk by the proposed method when it is computed with different number of samples?
> > >
> > >
> > > In the table below, we provide the 95% confidence interval for the importance sampling estimate with an increasing number of samples.
> > >
> > > | Model                  | Mean        | CI (n=200)                      | CI (n=500)                      | CI (n=1000)                     | CI (n=1500)                     |
> > > |------------------------|------------|----------------------------------|----------------------------------|----------------------------------|----------------------------------|
> > > | Qwen2.5-7B-Instruct    | 0.017376 | (0.009855, 0.025771)        | (0.009866, 0.025244)        | (0.00983, 0.025046)        | (0.009722, 0.025030)        |
> > > | Olmo-3-7B-Instruct     | 0.00141 | (0, 0.00330)                 | (0, 0.00300)                 | (0, 0.00384)                 | (0, 0.00342)                 |
> > > | Llama-3.1-8B-Instruct  | 0.01190 | (0.00473, 0.02009)        | (0.00496, 0.02051)        | (0.00462, 0.01899)        | (0.00458, 0.01922)        |
> > > | Llama-3.2-1B-Instruct  | 0.12585 | (0.09716, 0.13882)        | (0.09882, 0.14143)        | (0.09922, 0.13905)        | (0.10498, 0.14671)        |
> > > | phi-4                  | 0.00440 | (0.00084, 0.00639)        | (0.00099, 0.00709)        | (0.00132, 0.00738)        | (0.00140, 0.00740)        |

---

### Official Review · Reviewer_tUpv · 2026-03-11

**Soundness:** 3
**Presentation:** 3
**Significance:** 3
**Originality:** 3
**Overall Recommendation:** 4
**Confidence:** 5

**Summary:**

Due to safety finetuning essentially shaping the output distribution towards reducing the probability of harmful outputs, they can still surface rarely and their occurrence is not completely eliminated. The authors demonstrate this risk by sampling safety-tuned LLMs multiple times on queries from StrongREJECT and show that the fraction of queries with at least one harmful output increases as we increase the number of samples. Since naive sampling is prohibitively expensive, the authors theoretically motivate the use of importance sampling for  estimating rare event occurrence, introduce two refinement techniques to bring an unsteered unsafe model's use as the proposal model closer to the target model and optimize the hyperparameters($\lambda_{steer}$, $\alpha_{mix}$, and $t_{switch}$) using the cross-entropy method. The authors find that importance sampling coupled with their novel proposal selection methods converges faster than naive brute-force sampling at the same budget consequently also reaching the same estimates in 10-20x fewer samples. Beyond harmful outputs, the paper also presents tail risk estimation for other safety-relevant properties like sycophancy and hallucination, finding that Importance Sampling achieves equal or better performance to naive brute force estimation. Finally, the authors also demonstrate the existence of risks due to paraphrases of input prompts that can manifest as deployment risks from unseen queries and show that these can be approximated to a high degree using paraphrases of StrongREJECT input queries.

**Compliance With Llm Reviewing Policy:**

Affirmed.

**Final Justification:**

The authors engaged with my main concerns well addressing specific category divergence and other clarifications that reduce my objection with the use of activation steering. As a result, I've updated my significance score from 2 to 3 and overall score for 3 to 4.

**Key Questions For Authors:**

Q.1 How sensitive are the importance sampling estimates to the choice of datasets used to compute the steering vector?
Q.2 Are there categories of harm in StrongREJECT where the importance sampling estimates systematically diverge from brute-force estimates, and if so, does this correlate with the steering vector's coverage of those harm types?
Q.3 At what probability level do you expect the variance of the importance sampling estimator to become too large to be useful, even with the optimized proposal?

**Limitations:**

The authors do not discuss the following major limitations:
- The reliance on activation steering ties estimation reliability to a technique with known issues specifically in out-of-distribution performance
- The method's practical floor may be insufficient to estimate tail risks in frontier models, models that are arguably most capable at causing safety problems

**Strengths And Weaknesses:**

Strengths:
- The paper studies tail risks in LLMs and introduces Importance Sampling with an unsafe steered model as the proposal model to approximate the likelihood of harmful responses that subvert safety training. This is a fairly underexplored and timely research direction.
- The paper also demonstrates these tail risks well through harmful responses for StrongREJECT inputs and harmful responses for paraphrased inputs from StrongREJECT. I believe this paper can be quite useful as a reference purely for this demonstration.
- The proposed method is also shown to be able to provide strong estimates using 10-20x lower number of queries than naive brute-force sampling.
- The paper is overall well-written with easy-to-understand figures.

Weaknesses:
- The authors use activation steering to develop their proposal model making their proposal model significantly data-dependent. Tan et al.[1] have previously uncovered several issues with steering vectors, out-of-distribution brittleness being specifically of interest to this work. The use of activation steering thus significantly undermines the reliability of importance sampling on inputs that seem related to the dataset used to train the steering vector but are squarely out-of-distribution.
- The repeated sampling demonstration while useful, demonstrates tail risks for model smaller than 14B parameters. Figure 2 shows that for some models it can take upto $10^4$ samples to reach a ~10% harmful response rate, robustness to repeated sampling also seems to be correlated with scale and more recent models with likely more robust safety training. It is unclear whether Importance Sampling scales in efficiency as tail probabilities decrease calling the utility of the proposed method in question. Jones et al.[2] do show that risk from repeated sampling in frontier models goes up to $10^6$ to $10^8$ queries motivating further evidence that the proposed method does provide an efficiency increase at this scale.

Minor:
- The StrongREJECT judge is employed for harmfulness evaluation which has its own tail distribution and error rate making this risk potentially not as stark as shown in Figure 2. The StrongREJECT paper reports a 0.08 MAE and the curve in Figure 2 should account for this
- Typo: - Line 135: “Llama-3.2-8B provides harmful responses to fewer queries than Qwen2.5-7B”, type - Llama-3.1-8B

[1] Tan, D., Chanin, D., Lynch, A., Paige, B., Kanoulas, D., Garriga-Alonso, A., & Kirk, R. (2024). Analysing the generalisation and reliability of steering vectors. Advances in Neural Information Processing Systems, 37, 139179-139212.

[2] Jones, E., Tong, M., Mu, J., Mahfoud, M., Leike, J., Grosse, R., ... & Sharma, M. (2025). Forecasting rare language model behaviors. arXiv preprint arXiv:2502.16797.

---

> ### Author Rebuttal · Authors · 2026-03-30
>
> **Thank you for your feedback. We deeply appreciate your positivity about our work! We have added clarifications below, and believe the added context is sufficient to increase your overall score.**
>
> *> The authors use activation steering [...] Tan et al.[1] have previously uncovered several issues with steering vectors, out-of-distribution brittleness...*
>
> Our work actually shows that an imperfect steering method (activation steering) can be used for unbiased (perfect) estimation when combined with classical variance reduction methods (i.e. importance sampling) to yield sample-efficient, unbiased, and accurate estimates! (See Fig. 3, 4, 5)
>
> *> The use of activation steering thus significantly undermines the reliability of importance sampling on inputs that [...] are squarely out-of-distribution.*
>
> If activation steering yields low-quality proposals, the CEM search method reverts to the target model, as it is always included in the CEM search procedure. However, in our experiments CEM always selects non-zero steering intensity.
>
> We use off-the-shelf steering vectors from prior work as a training-free method for constructing proposal models.  Proposal selection can be further improved by including additional families (e.g. prompting, SFT, and RL) in the search space. Hence, our results are a methodological lower bound on how well we can do rare-event estimation for language models.
>
> *> How sensitive are the importance sampling estimates to the choice of datasets used to compute the steering vector?*
>
> We use off-the-shelf steering vectors. Future work could explore how to construct better steering vectors, as well as alternative proposal families.
>
> There are likely cases where activation steering would not significantly improve on MC estimates, and the alternatives above would be useful.
>
> *> Are there categories of harm in StrongsREJECT where the importance sampling estimates systematically diverge?*
>
> We include the requested error results below. There is no single category across all models where importance sampling estimates systematically diverge most.
>
> | Category | Qwen2.5-7B | Olmo-3-7B | Llama-3.1-8B | Llama-3.2-1B | Phi-4 |
> |---|---|---|---|---|---|
> | Disinfo & deception | 1.12e-3 | 7.07e-5 | 9.59e-4 | 2.64e-3 | 1.99e-3 |
> | Hate | 4.50e-4 | 1.58e-4 | 1.28e-4 | 4.08e-3 | 5.04e-5 |
> | Illegal goods | 2.15e-4 | 5.58e-5 | 7.72e-4 | 5.26e-3 | 1.06e-4 |
> | Non-violent crimes | 5.32e-4 | 1.40e-3 | 1.74e-3 | 1.75e-3 | 9.38e-4 |
> | Sexual | 1.48e-4 | 1.25e-5 | 6.96e-5 | 2.19e-3 | 1.78e-4 |
> | Violence | 8.54e-4 | 1.56e-5 | 8.63e-4 | 7.85e-4 | 9.92e-4 |
>
> *> **W2 & Q3** : [...] It is unclear whether Importance Sampling scales in efficiency as tail probabilities decrease calling the utility of the proposed method in question. [...] motivating further evidence that the proposed method does provide an efficiency increase at this scale….At what probability level do you expect the variance of the importance sampling estimator to become too large to be useful, even with the optimized proposal?*
>
> ***This reinforces the necessity of our approach!*** Efficient estimation methods like importance sampling become increasingly important as harmful events become rarer, and we observe this trend across experiments (See Fig. 3, 5).  What may change is the number of required samples necessary for an accurate estimate. Approaches such as adaptive importance sampling could automate sample size selection.
>
> *> The method's practical floor may be insufficient to estimate tail risks in frontier models…*
>
> Respectfully, we do not agree with the speculation that our method’s “practical floor” may be insufficient for frontier models.  As a general trend, ***we find that the rarer an event is, the more importance sampling outperforms standard, naive Monte Carlo estimation.***
>
> If you are concerned about constructing a proposal model for frontier models, there is a swath of literature on manipulating and training models to misbehave. Our method could be combined with proposal models learned by RL or SFT to up-sample the trait being evaluated. Moreover, most frontier labs have helpful-only models that can be prompted and fine-tuned to exhibit undesirable behaviors that we would like to measure.
>
> *> The StrongREJECT judge is employed for harmfulness evaluation which has its own tail distribution…*
>
> This is an excellent observation. As we acknowledge in the Limitations section, our framework requires an accurate judge. We are excited about future work that improves judge capabilities. Additionally, advanced judges that require additional compute overhead will further increase the cost of evaluation, and further motivate our sample-efficient estimation method!
>
> **Please let us know if you have any other questions that we can address. If not, we sincerely hope you can consider increasing your rating to an Accept, as we believe that our contribution is significant to the safety community that aims to increase the credibility of evaluation techniques.**

---

> > ### Author Rebuttal · Reviewer_tUpv · 2026-04-02
> >
> > I thank the reviewers for their extended experiments and clarifications. I'll update my score to a 4.

---

> > > ### Author Response · Authors · 2026-04-03
> > >
> > > Given that you believe our work addresses an “underexplored and timely research direction,” and you found our paper to be “well-written” and “quite useful as a reference” are there any additional questions or clarifications we can address that will increase your score to a full accept?

---

### Official Review · Reviewer_X2de · 2026-03-13

**Soundness:** 2
**Presentation:** 2
**Significance:** 3
**Originality:** 2
**Overall Recommendation:** 4
**Confidence:** 4

**Summary:**

This paper proposes a method to estimate the probability that a language model produces rare harmful outputs given the same input. To avoid brute-force sampling, the authors use activation steering to construct a more unsafe proposal model, and then combine it with importance sampling to estimate such extremely low-probability events more efficiently.

**Compliance With Llm Reviewing Policy:**

Affirmed.

**Final Justification:**

The authors’ response has addressed most of my concerns. Overall, I believe this paper is worthy of acceptance, and therefore I have raised my score from 3 to 4.

**Key Questions For Authors:**

1. How do the authors ensure that the ground truth of the rare harmful output probability in this paper is truly correct?
2. What exactly is estimated in this paper: the true harmfulness probability, or the probability of being judged as harmful by some judge?
3. How do other modules or hyperparameters, such as the mixing and switching hyperparameters, affect the results?
4. Is activation steering necessary for constructing the proposal model? Are there simpler and more feasible alternatives?

**Limitations:**

yes

**Strengths And Weaknesses:**

**Strengths**

1. This paper proposes a method to evaluate the probability that any input query causes a large language model to produce harmful outputs, and effectively improves the current neglect of the probabilistic nature of the model and tail output behaviors in evaluation.
2. Through experiments on benchmarks measuring misuse and misalignment, the proposed evaluation method can achieve effects comparable to brute-force Monte Carlo estimation with fewer samples.

**Weaknesses**

1. The related work of this paper is insufficient. Besides the earlier works [1,2] mentioned in the paper, there are actually many papers related to estimating harmful output probability and rare harmful outputs, such as the recent [3,4,5,6]. The authors did not sufficiently discuss the related work, which limits the novelty. The authors should even compare with related methods in the experiments.
2. The method in this paper contains many heuristic designs, such as the Activation Steering module and the Model Switching module. Too many intuition-based designs make the theoretical and practical feasibility difficult to guarantee. Therefore, providing code is a basic prerequisite for demonstrating reproducibility, but the authors overlooked this point.
3. This paper should include an algorithm framework figure to show the whole pipeline more clearly. In addition, it should also provide algorithm pseudocode. This is very important.
4. Conventional harmful output probability estimation can be done with black-box access, while this paper is based on activation steering and requires white-box access, which limits its applicability to some extent. Moreover, the validation of this method is mainly based on STRONGREJECT, which makes its generalization in real use questionable. In addition, the ablation study is also insufficient.

[1] Jones, Erik, Meg Tong, Jesse Mu, Mohammed Mahfoud, Jan Leike, Roger Grosse, Jared Kaplan, William Fithian, Ethan Perez, and Mrinank Sharma. "Forecasting rare language model behaviors." *arXiv preprint arXiv:2502.16797* (2025).

[2] Wu, Gabriel, and Jacob Hilton. "Estimating the probabilities of rare outputs in language models." *arXiv preprint arXiv:2410.13211* (2024).

[3] Broadwater, Keita. "Evaluating LLM Safety Under Repeated Inference via Accelerated Prompt Stress Testing." *arXiv preprint arXiv:2602.11786* (2026).

[4] McAllister Dorman, Jake, Edward Gillman, Dominic C. Rose, Jamie F. Mair, and Juan P. Garrahan. "Rare Event Analysis of Large Language Models." *arXiv e-prints* (2026): arXiv-2602.

[5] Feng, Mingqian, Xiaodong Liu, Weiwei Yang, Chenliang Xu, Christopher White, and Jianfeng Gao. "Statistical Estimation of Adversarial Risk in Large Language Models under Best-of-N Sampling." *arXiv preprint arXiv:2601.22636* (2026).

[6] Xu, Wenzhuo, Zhipeng Wei, Xiongtao Sun, Zonghao Ying, Deyue Zhang, Dongdong Yang, Xiangzheng Zhang, and Quanchen Zou. "Probabilistic Modeling of Jailbreak on Multimodal LLMs: From Quantification to Application." *arXiv preprint arXiv:2503.06989* (2025).

---

> ### Author Rebuttal · Authors · 2026-03-30
>
> **Thank you for your feedback. We have added clarifications below, and hope the added context is sufficient to increase your score.**
>
> *> The related work of this paper is insufficient…*
>
> Thank you for highlighting these works! [3], [4], and [5] were all published **after the ICML 2026 submission deadline**, so it would have been impossible for them to be discussed.
>
> [6] is quite different from our work, but still worth mentioning. They are concerned with jailbreaking multi-modal models. They learn a jailbreak success probability prediction network which they obtain by doing Monte Carlo sampling on a set of up to 40 samples per query. **Our results demonstrate that 40 samples is insufficient to elicit several rare behaviors,** and our proposed method could **further improve** the training of their regressor.
>
> We have included a discussion of these papers in the revised draft.
>
> *> The method in this paper contains many heuristic designs, such as the Activation Steering module and the Model Switching*
>
> Model switching and mixing are standard tools (see [1]) used to reduce the variance of importance weights. See our responses to reviewers Eutt and tUpv for a discussion for how combining activation steering and importance sampling yields an unbiased and consistent estimator.
>
> [1] Owen, A. and Zhou, Y. Safe and effective importance sampling. *Journal of the American Statistical Association*, 2000
>
> *> Providing code is a basic prerequisite for demonstrating reproducibility.*
>
> We release our code here: https://anonymous.4open.science/r/rare-generations-D42B/
>
> *> This paper should include an algorithm framework figure to show the whole pipeline more clearly. In addition, it should also provide algorithm pseudocode.*
>
> Our current version has a description of the full estimation pipeline on pages 4-5 (lines 211-219 and 250-257). We have added algorithm pseudocode in our revised draft.
>
> *> Conventional harmful output probability estimation can be done with black-box access, while this paper is based on activation steering and requires white-box access, which limits its applicability to some extent.*
>
> While our method does require white-box access, the conventional approach for estimating harmful output probabilities with black-box access is repeated sampling, which cannot be used to efficiently estimate rare events (see Figure 2 and Table 1). We instead provide a method that uses white-box access for efficient estimation. *The primary use-case for our method is testing model risk pre-deployment, where white-box access is generally available.*
>
> *> Moreover, the validation of this method is mainly based on STRONGREJECT, which makes its generalization in real use questionable.*
>
> We use two more misalignment datasets in Section 5, introduced by the paper studying persona vectors (Chen et al. 2025), with different judges to show that our method generalizes to other scenarios.
>
> *> In addition, the ablation study is also insufficient.*
>
> Which ablation study are you referring to? We are happy to provide additional experimental results and information.
>
> *> How do the authors ensure that the ground truth of the rare harmful output probability .. is truly correct?*
>
> A challenge in rare-event estimation is that, typically, there is no way to compute ground truth probabilities, barring toy settings.
>
> We use brute-force Monte Carlo estimates (10,000 samples per prompt) as a pseudo ground truth to gauge how well our importance sampling approach matches more expensive estimates. In several cases, the importance sampling estimate is > 0.0, but the MC estimate is 0.0, indicating that the probability of harmful outcomes is less than 1/10,000. Therefore a faithful MC estimate requires more than 10,000 samples.
>
> *> What exactly is estimated in this paper: the true harmfulness probability, or the probability of being judged as harmful by some judge?*
>
> The probability of the output being judged as harmful by a fixed judge. See Equation 1. We note in the Limitations section that LLM-as-judges are fallible, and are excited for complementary work that improves their reliability.
>
> *> How do other modules or hyperparameters, such as the mixing and switching hyperparameters, affect the results?*
>
> The choice of hyperparameters affects the variance of the estimate. Several choices of hyperparameters provide high-quality estimates. In Figure 8, we show that the CEM selected proposal model reduces estimator variance and is critical to providing efficient and low-variance estimation.
>
> *> Is activation steering necessary for constructing the proposal model?*
>
> See our response to Reviewer [tUpv](https://openreview.net/forum?id=Joka19sTny&noteId=RJpFtyE0kC).
>
> **Please let us know if you have any other questions that we can address. If not, we sincerely hope you can consider increasing your rating to an Accept, as we believe that our contribution is significant to the safety community that aims to increase the credibility of evaluation techniques.**

---

> > ### Author Rebuttal · Reviewer_X2de · 2026-04-03
> >
> > Thank you for the authors’ response. It has addressed some of my concerns. However, several issues still warrant clarification:
> >
> > 1. How necessary is a heuristic design such as activation steering in this setting? For example, would more conventional alternatives, such as feature-based methods or logit-level modifications, already be sufficient?
> >
> > 2. Given the Monte Carlo estimation procedure and the LLM-as-a-judge setup described by the authors, how can one ensure that the resulting “ground truth” is in fact correct in a factual sense?

---

> > > ### Author Response · Authors · 2026-04-03
> > >
> > > > *1. How necessary is a heuristic design such as activation steering in this setting? For example, would more conventional alternatives, such as feature-based methods or logit-level modifications, already be sufficient?*
> > >
> > > Activation steering is a very popular method for controlling and studying model behavior (e.g. see Claude Opus 4.6 system card), with a large amount of off-the-shelf support. Future work on controllability and steering could explore alternative ways of controlling model behavior, which could easily be operationalized within our framework.
> > >
> > > Additionally, we further explain the choice of activation steering in our rebuttal to Reviewer tUpv:
> > >
> > > >> We use off-the-shelf steering vectors from prior work as a training-free method for constructing proposal models.  Proposal selection can be further improved by including additional families (e.g. prompting, SFT, and RL) in the search space. Therefore, our results are a methodological lower bound on how well we can do rare-event estimation for language models.
> > >
> > > **Follow-up questions:**
> > >
> > > 1. Activation steering is the most conventional feature-based method we are aware of for steering LLMs. Are there alternative feature-based methods that you would like us to discuss?
> > > 2. For clarification, we also do use logit-level modifications, specifically model mixing (a commonly used tool in sequential importance sampling, see eq. 7 in [1]) and model switching. We discuss both these modifications in the **main text** in Section 3.2 from lines 201 onwards. Would you like us to consider other types of logit-level modifications?
> > >
> > > Please let us know if there are any **specific** non-heuristic feature-based or logit-level modifications that you think we should consider. We hope the clarifications above are sufficient for you to increase your score.
> > >
> > > [1] Owen, Art, and Yi Zhou. "Safe and effective importance sampling." Journal of the American Statistical Association 95.449 (2000): 135-143.
> > >
> > > > *2. Given the Monte Carlo estimation procedure and the LLM-as-a-judge setup described by the authors, how can one ensure that the resulting “ground truth” is in fact correct in a factual sense?*
> > >
> > > Respectfully, we answered this question in the rebuttal:
> > >
> > > >> A challenge in rare-event estimation is that, typically, there is no way to compute ground truth probabilities, barring toy settings with analytical probabilities.
> > >
> > > >> We use brute-force Monte Carlo estimates (10,000 samples per prompt) as a pseudo ground truth to gauge how well our importance sampling approach matches more expensive estimates. In several cases, the importance sampling estimate is > 0.0, but the MC estimate is 0.0, indicating that the probability of harmful outcomes is less than 1/10,000. Therefore a faithful MC estimate requires more than 10,000 samples.
> > >
> > > >> The probability of the output being judged as harmful by a fixed judge. See Equation 1. We note in the Limitations section that LLM-as-judges are fallible, and are excited for complementary work that improves their reliability.
> > >
> > > We believe large-scale automated evaluations will inevitably suffer from these limitations. While our work mitigates the need for brute-force estimates, relying on LLMs-as-judge (or a BERT-like classifier) is necessary for automated evaluations.
> > >
> > > Please let us know if you need any further clarification.

---

### Decision · Program_Chairs · 2026-04-30

**Decision:**

Accept (spotlight)

**Comment:**

After the rebuttal phase, this paper maintains four accepts from the four reviewers. The reviewers noted that this paper includes thorough experiments, is well-written and easy to understand, addresses an under-explored problem, and proposes and intuitive method. The rebuttal phase was also productive: The authors addressed many of the concerns raised by the reviewers, which was noted by the reviewers toward the end of the rebuttal period.  Given the novelty of the paper and the consensus of the reviewers, this is a clear accept in my book.